# Extracellular Prostaglandins E1 and E2 and Inflammatory Cytokines Are Regulated by the Senescence Program in Potentially Premalignant Oral Keratinocytes

**DOI:** 10.3390/cancers14112636

**Published:** 2022-05-26

**Authors:** Lee Peng Karen-Ng, Usama Sharif Ahmad, Luis Gomes, Keith David Hunter, Hong Wan, Eleni Hagi-Pavli, Eric Kenneth Parkinson

**Affiliations:** 1Center for Oral Immunobiology and Regenerative Medicine, Institute of Dentistry, Barts and The London School of Medicine and Dentistry, Queen Mary University of London, Turner Street, London E1 2AD, UK; karennlp@um.edu.my (L.P.K.-N.); u.s.ahmad@qmul.ac.uk (U.S.A.); luis.mgs.gomes@gmail.com (L.G.); h.wan@qmul.ac.uk (H.W.); e.hagi-pavli@qmul.ac.uk (E.H.-P.); 2Oral Cancer Research & Coordinating Center (OCRCC), Faculty of Dentistry, Universiti Malaya, Kuala Lumpur 50603, Malaysia; 3Academic Unit of Oral and Maxillofacial Medicine and Pathology, School of Clinical Dentistry, University of Sheffield, Sheffield S10 2TA, UK; keith.hunter@liverpool.ac.uk; 4Liverpool Head and Neck Centre, Molecular and Clinical Medicine, University of Liverpool, Liverpool L1 8JX, UK

**Keywords:** oral, pre-malignancy, keratinocyte, cancer, senescence, p53, prostaglandin, p38 mitogen-activated kinase, cyclo-oxygenase, ataxia telangiectasia mutated, telomerase

## Abstract

**Simple Summary:**

The early treatment of oral cancer is a high priority, as improvements in this area could lead to greater cure rates and reduced disability due to extensive surgery. Oral cancer is very difficult to detect in over 70% of cases as it develops unseen until quite advanced, sometimes rapidly. It has become apparent that there are at least two types of epithelial cells (keratinocytes) found in oral tissue on the road to cancer (premalignant). One type secretes molecules called prostaglandins but the other does not and the former may stimulate the latter to progress to malignancy, either by stimulating their proliferation or encouraging the influx of blood vessels to feed them. Additionally, we have identified regulators of prostaglandin secretion in premalignant oral cells that could be targeted in future therapies, such as inducers of cellular senescence, drugs which kill senescent cells (senolytics), steroid metabolism, cyclo-oxygenase 2 (COX2) and p38 mitogen-activated protein kinase.

**Abstract:**

Potentially pre-malignant oral lesions (PPOLs) are composed of keratinocytes that are either mortal (MPPOL) or immortal (IPPOL) in vitro. We report here that MPPOL, but not generally IPPOL, keratinocytes upregulate various extracellular tumor-promoting cytokines (interleukins 6 and 8) and prostaglandins E1 (ePGE1) and E2 (ePGE2) relative to normal oral keratinocytes (NOKs). ePGE upregulation in MPPOL was independent of PGE receptor status and was associated with some but not all markers of cellular senescence. Nevertheless, ePGE upregulation was dependent on the senescence program, cyclo-oxygenase 2 (COX2) and p38 mitogen-activated protein kinase and was partially regulated by hydrocortisone. Following senescence in the absence of p16^INK4A^, ePGEs accumulated in parallel with a subset of tumor promoting cytokine and metalloproteinase (MMP) transcripts, all of which were ablated by ectopic telomerase. Surprisingly, ataxia telangiectasia mutated (ATM) function was not required for ePGE upregulation and was increased in expression in IPPOL keratinocytes in line with its recently reported role in telomerase function. Only ePGE1 was dependent on p53 function, suggesting that ePGEs 1 and 2 are regulated differently in oral keratinocytes. We show here that ePGE2 stimulates IPPOL keratinocyte proliferation in vitro. Therefore, we propose that MPPOL keratinocytes promote the progression of IPPOL to oral SCC in a pre-cancerous field by supplying PGEs, interleukins and MMPs in a paracrine manner. Our results suggest that the therapeutic targeting of COX-2 might be enhanced by strategies that target keratinocyte senescence.

## 1. Introduction

Oral cancer, predominantly oral squamous cell carcinoma (OSCC), is the sixth most common cancer worldwide and the management of this cancer has barely improved in decades [1]. One of the problems in treating OSCC is that they frequently develop from a field of genetically and phenotypically diverse cells that are clinically and sometimes histologically undetectable [2]. Advanced cancers, and even some high-risk potentially pre-malignant lesions (PPOLs), have extensive gene copy number variations [3] and develop genetic heterogeneity and consequently a platform for drug resistance [4]. Recent evidence suggests that this genetic heterogeneity [5] and the induction of cellular senescence [6] develops prior to most OSCCs becoming clinically identifiable [5] and immortal keratinocytes with extensive gene copy number variations have been demonstrated in PPOLs of a low histological grade [7].

It has long been known that even advanced OSCCs contain both mortal and immortal keratinocytes [8,9] and the same is true for PPOLs [7,10]. Moreover, despite the fact that both mortal (MPPOL) and immortal (IPPOL) keratinocytes have neoplastic-like phenotypes [8], altered transcriptional [10] and metabolic [11] profiles, MPPOLs have no gene copy number variations, gene methylation and few classical ‘driver’ mutations [12]. Similar keratinocytes have been detected in OSCC [7,8,10,12]. MPPOLs and IPPOLs may often be completely distinct lesions as their transcriptional profiles appear to diverge upon progression to mortal and immortal OSCC keratinocytes, although in some instances MPPOLs may be precursor lesions of IPPOLs [13,14,15]. In addition, MPPOL and MOSCC are associated with a different class of cancer-associated fibroblasts [16]. Whilst a considerable amount is known about cancer-associated fibroblasts and their role in modulating carcinoma behaviour, including OSCC [17], far less is known about the different types of keratinocytes that exist within cancerous and pre-cancerous fields and how they may influence each other’s behaviour.

Senescence can be induced by a variety of cellular stresses but is bypassed in normal keratinocytes by the dual reduction of p53 and p16^INK4A^ function and telomerase deregulation [18]. Genetic alterations in all of these pathways are common in OSCCs in vivo [19] and in immortal cell lines derived from PPOL [7] and OSCC [20]. In addition, under the conditions we have used here, keratinocyte senescence and differentiation can be reversibly bypassed by treating cells with the Rho-activated kinase (ROCK) inhibitor Y27632 (ROCKi) without changing karyotype or the DNA damage response [21]. 

Senescence is induced by certain oncogenes [22], yet not others [23], and this is sometimes referred to as oncogene-induced senescence (OIS). There is evidence suggesting that to overcome OIS in vitro, an intracrine function involving interleukin 6 (IL-6) and interleukin 8 (IL-8) must be inactivated [24]. The cytokine network is orchestrated in both normal and OIS by interleukin-1 alpha (IL-1α) [25] and is responsible for the clearance of pre-malignant cells in vivo in concert with the innate and adaptive immune systems [26].

We recently conducted an unbiased metabolomic screen of the extracellular metabolites of normal, MPPOL and IPPOL keratinocytes and identified extracellular elevation of a number of metabolites in MPPOL, including extracellular prostaglandins E1 and E2 (ePGEs1 and 2 respectively) and their related metabolites [11]. PGEs 1 and 2 are generated from arachidonic acid and gamma dihomo-linolenate by the enzymes cyclooxygenases 1 (COX1) and 2 (COX2), which are the established drug targets of the selective non-steroidal anti-inflammatory drug (NSAID), celecoxib [27,28]. However, the source of PGE1 and PGE2 in what is a complex mixture of cells is difficult to ascertain.

We have confirmed the results of the metabolic screen in that ePGEs 1 and 2 are strikingly upregulated in most MPPOL keratinocytes but in very few of the IPPOL keratinocytes. The aim of the present study was to investigate the regulatory mechanisms underpinning the earlier observations and in particular to identify new approaches to oral cancer prevention and/or therapy. We provide evidence that ePGE upregulation is dependent on the keratinocyte senescence program and pathways known to regulate the senescence associated secretory phenotype (SASP). We also show that exogenous ePGE2 increases IPPOL keratinocyte proliferation at physiologically relevant doses. We conclude that ePGEs 1 and 2, in conjunction with SASP cytokines [10,29], may contribute to IPPOL progression in a paracrine manner and suggest that targeting of these keratinocytes by anti-senescence therapies might be a novel approach to the management of PPOL, field cancerisation and OSCC. 

## 2. Materials and Methods

### 2.1. Cell Culture

The normal, ATM−/− and PPOL human keratinocytes were cultivated using lethally irradiated 3T3 feeders as described previously [7,8]. Briefly, keratinocytes were cultured in Dulbecco’s Modified Eagles Medium (DMEM) containing 10% vol/vol fetal bovine serum (FBS–Hyclone foetal clone II) supplemented with 0.4 μg/mL hydrocortisone (HC), 2 mM glutamine ((GE Healthcare, PAA Laboratories GmbH, Pasching, Austria) and antibiotics (streptomycin 50 U/mL Penicillin and 50 µg/mL Streptomycin, Sigma, Poole, Dorset, UK) in an atmosphere of 10% CO_2_/90% air. Keratinocytes were sub-cultured once weekly at a density of 1 × 10^5^ cells per 9 cm plate to prohibit confluence. Human oral fibroblasts were cultured similarly in DMEM without HC.

### 2.2. Cell Lines Used in the Study

The characteristics and clinical details of the M- and IPPOL lines and patient donors used in the study have been published previously [7,10,11,15]. Immortal normal oral keratinocytes from the floor of the mouth, OKF6/TERT-1 (OKF6) [14], OKF4/CDK4R/P53DD/TERT (OKF4) [18] were a generous were gift of James Rheinwald (Brigham and Women’s Hospital Boston U.S.A.) and have been described previously. The normal gingival keratinocytes (NHOK810) were a generous gift of Angela Hague (University of Bristol, Bristol, UK) and have been described previously [11]. The Ataxia Telangiectasia Mutated (ATM−/−) epidermal keratinocytes were obtained from a foreskin of patient AT5BI [30]. The H157 keratinocytes [31] were a generous gift of Professor Stephen Prime (University of Bristol, UK), HeLa cells were a generous gift Professor Margaret Stanley (University of Cambridge, Cambridge, UK) and have been characterized [32,33], the HEK293 Phoenix A cells were obtained from Nolan Laboratory, Stanford, CA and the HCA/C29 (HCA-7 colony 29) cells were obtained from Sigma-Aldrich cat# 02091238. Normal human epidermal keratinocytes NHEK92 and NHEK131 were obtained from Gibco Thermo-Fisher scientific UK and consisted of two batches (batch #92 and batch #131, respectively) each of which was derived from 3–6 donors. Normal human fibroblast line NHOF-1 [34] was obtained from a biopsy of buccal mucosa.

The details of the number of lines analyzed (*n*) are given in the figure legends, but ranged from between two and four in the case of MPPOL and between two and eight in the case of IPPOL. The different classes of cell lines were color-coded as indicated.

### 2.3. Collection of the Conditioned Medium and Cell Pellets for Analysis of the Extracellular Metabolites and Proteins

The collection of the conditioned medium and cell pellets was carried out as described previously [35,36] except that the keratinocytes were allowed to reach confluence before the collection began to more accurately represent the state of the epithelium in vivo. 3T3 feeders were removed as described previously [7,8]. The keratinocytes were plated in such a way as to reach confluence within 72 h in T25 flasks before adding 3 mL fresh medium to each flask for 24 h and snap freezing the fresh medium at zero time to elucidate the background [35,36]. To achieve confluence in 72 h, the cells were disaggregated, trypsin/EDTA neutralized, cells were counted on a at hhaemocytometer, pelleted by centrifugation at 300 g and the amount of protein in each cell pellet was measured and the results expressed in pg or ng/mL medium/mg cell protein. The cell densities plated to achieve confluence have been reported [11] (see also Appendix A).

Total cell protein concentration was measured using the DC Protein Assay kit (Bio-Rad Laboratories, Hertfordshire, UK) and the results were expressed as amount/mL/mg cell protein. In some experiments, the 3T3s were removed and the keratinocytes were allowed to reach confluence on the 3T3-deposited matrix and the data were expressed as pg/mL/10^5^ cells or pg or ng/mL medium/mg cell protein. This method yielded more reproducible data but did not affect the pattern of the results.

### 2.4. Enzyme-Linked Immunosorbent Assay (ELISA)

Two types of ELISA were used to measure the SASP cytokines released from the cells. To measure PGE1 and PGE2, a competitive ELISA technique was utilized (Abcam, Cambridge, UK) and for the cytokines (IL-1α, IL-1β, IL-6 and IL-8), a sandwich ELISA method (Quantikine^®^ ELISA Immunoassay, R&D Systems, Abingdon, UK) was employed to measure these proteins. The manufacturer’s protocol was followed in both ELISA types. The detection limits for PGE1 and PGE2 are 4.88–5000 pg/mL and 39.1–2500 pg/mL respectively. The detection limits of the Quantikine^®^ ELISA Immunoassay kits were 3.9–250 pg/mL IL-1α/β; 3.13–300 pg/mL IL-6 and 3.13–2000 pg/mL IL-8. 

### 2.5. Knockdown of p53, COX-1 and COX-2 in OKF6

To deliver the p53/COX-1/COX-2 siRNA into OKF6 cells, Dharmacon DharmaFECT 1 transfection reagent (GE Healthcare, Manchester, UK) was used following the manufacturer’s siRNA transfection protocol with slight modification. Each experiment was performed in triplicate with the following samples: untreated cells (mock), test siRNA (p53/COX-1/COX-2) and negative control siRNA (non-targeting/scramble). 

When siRNA is processed by the RNA-Induced Silencing Complex (RISC), off-target effects may occur and the downregulation of unintended targets occurs. Off-target activity changes in gene expression which complicates the interpretation of phenotypic effects in gene-silencing experiments and can potentially lead to unwanted toxicities [37]. To avoid off-target effect activity, pooling of multiple siRNAs is essential as previously described [37]. In this study, ON-TARGETplus Human TP53/PTGS1/PTGS2 siRNA SMARTpool by Dharmacon (GE Healthcare, Manchester, UK) was used. The target sequences of the siRNAs were detailed as below in Appendix A.

### 2.6. Western Blotting

Cell pellets were thawed on ice before adding 50–100 µL radio immunoprecipitation assay (RIPA) buffer (Sigma, Dorset, UK) with protease and phosphatase inhibitors (Roche, Hertfordshire, UK) and were lysed for 30 min at 4 °C. In some experiments, lysis buffer was added directly to the plate. Following removal of cell debris by centrifugation at 12,000 rpm for 20 min at 4 °C, the protein quantitated by the DC Protein Assay and total cellular protein was separated based on molecular weight on 4–12% gradient SDS sodium dodecyl sulphate polyacrylamide gels under denaturing and reducing conditions. Following protein transfer, the nitrocellulose membrane was blocked with 5% weight/vol milk protein prepared in Tris Buffer Saline and Tween 20 (TBS-T) for 1 h at room temperature (RT). The primary antibodies were diluted in 5% wt/vol milk protein in TBS-T and the membrane/blot was incubated/probed overnight with primary antibody at 4 °C, washed 3 times in TBS-T for 5 min at RT under agitation. The membranes were incubated with appropriate immunoglobulin G horseradish peroxidase (IgG HRP)—conjugated secondary antibody, and diluted as above for 1 h at RT. Antigen-antibody complexes were detected by incubating with ECL Western Blotting Substrate for 1 min or for sensitive detection, ECL Prime Western Blotting Detection Reagent or SuperSignal^®^ West Femto Maximum Sensitivity Substrate for 5 min according to manufacturer’s protocols. Membranes were exposed to the Amersham Hyperfilm ECL and were developed in the dark using a standard film developer machine. Densitometry analysis was performed on scanned films using Image J. The relative intensities of the bands of interest were normalized against the values obtained from the corresponding loading controls. The primary antibodies, secondary antibodies, the suppliers, and the positive and negative controls are given in the Appendix A. The whole western blots were shown in Appendix A.

### 2.7. Metabolomic Analysis, Normalisation and Data Presentation as Scaled Intensity

The details of the metabolomics analysis have been published previously [35,36]. See Appendix A for details.

### 2.8. Drug Treatment of the Keratinocyte Cultures

Salicylic acid (Sigma-Aldrich, Dorset, UK) and Y-27632 (ROCKi; Enzo Life Sciences, Exeter, UK) were dissolved in growth medium or double distilled water, filter sterilized and used at the stated dose. SB203580 (Tocris Chemical Co., Bristol, UK) and KU55933 (R&D systems, Abingdon, UK), NS398 (Santa Cruz Biotechnology, Dallas, TX, USA) and celecoxib (Sigma-Aldrich, Dorset, UK) were dissolved in DMSO and used at the stated doses. The final concentration of DMSO was 0.1% vol/vol and DMSO alone was used as the vehicle control. PGE1 and PGE2 were purchased from Cayman Chemical (Michigan, USA). PGE1 and PGE2 were dissolved in 100% ethanol (Sigma-Aldrich, Dorset, UK) as a stock solution of 50 mg/mL and 100 mg/mL respectively and further diluted in 100% ethanol to the desired concentration. 0.1% Ethanol was used as the vehicle control. The function of Y-27632 was confirmed by Western blotting of the following antigens: NOTCH1 (ab87982) and Integrin α6 (ab181551) (Abcam, Cambridge, UK).

### 2.9. Statistical Analysis

A Student’s unpaired two-sided *t*-test was used to test the statistical difference between two data sets of even numbers, a Welch’s *t* test to test the difference between two data sets of uneven numbers and one-way ANOVA was used to test the difference between multiple data sets. A *p* value of <0.05 was considered significant.

## 3. Results

### 3.1. Extracellular PGEs 1 and 2 (ePGEs 1 and 2) Are Upregulated in MPPOLs

We confirmed and extended our observations obtained from the unbiased metabolomic screen to a larger panel of cell lines using ELISA assays for ePGEs 1 and 2. Three out of four MPPOL lines (D6, D30 and D25) had elevated levels of both ePGEs relative to the NOKs, which reached significance for ePGE1. However, six out of seven of the IPPOL lines (Figure 1a) and D17 (Figure 1b) did not over-express ePGEs, with the exception of line D4, which expresses one wild type p53 allele. D17 is interesting because although it has lost p16^INK4A^ expression [15] and possesses a *NOTCH1* mutation [11], it retains a wild type and functional p53 protein and regulates telomerase normally [38]. This pattern was independent of oral site (Figure 1b) as all these lines were from non-tongue sites. These data suggest that the breakdown of senescence leads to a loss of expression of the ePGEs.

### 3.2. ePGEs 1 and 2 Are Regulated by Senescence and Its Breakdown

#### 3.2.1. Ablation of Senescence with the Rho-Activated Kinase Inhibitor Y 27632 (ROCKi) Abolishes ePGE Expression in MPPOL Keratinocytes

We tested the levels of the senescence marker, senescence-associated beta galactosidase (SA-βGal), in the cell line panel (Appendix A). The MPPOL lines and senescent NOKs had higher levels of SA-βGal-positive cells than early passage NHOK810 keratinocytes, especially in the centers of keratinocyte colonies where the cornified and late terminally differentiated cells are located. However, the mid lifespan MPPOL line D17, which does not express p16^INK4A^ [7], and the immortal lines D19, DOK and OKF6/TERT-1 (OKF6) did not show elevated levels of SA-βGal-positive cells. We used three independent strategies to assess the role of the senescence program in the regulation of the ePGEs. Firstly, to address the role of senescence in the upregulation of the ePGEs in D6 and D30 MPPOLs we immortalized the cells by treating them with ROCK inhibitor Y-27632 (ROCKi), which has been reported to immortalize keratinocytes without changing their genotype [21] and then measuring the levels of the PGEs and SA-βGal. Figure 2a,b show that culturing the keratinocytes for 2 weeks with ROCKi eliminated the SA-βGal in the center of D30 colonies but had no effect on IPPOL line D19. Ablating the senescence program in D6 and D30 reduced the level of both ePGE1 (Figure 2c,d) and ePGE2 (Figure 2e,f) to the level of normal keratinocytes and the IPPOLs did not show any changes in the last two groups. In a separate series of experiments where the cultures were allowed to grow to confluence on 3T3 extracellular matrix, ROCKi dramatically reduced both ePGEs in D30, NOKs and D17, which were all capable of senescence but had little effect on the PGE levels of the immortal lines D19 and DOK. ROCKi also had a much reduced effect on OKF6 which, despite being immortal and expressing telomerase, did express considerable levels of both ePGEs, indicating that ROCKi generally only affected ePGE levels if the senescence program was intact (Figure 2g–j). 

#### 3.2.2. Ablation of Replicative Senescence with ROCKi Abolishes ePGE Expression in p16^INK4A^-deficient Line D17 

Next, we cultured the p16^INK4A^-deficient line D17 to the point where it approached senescence and compared its extracellular PGE levels with the same line cultured with ROCKi or the ectopic expression of the catalytic subunit of telomerase, *TERT*. Senescence was monitored by SA-βGal staining as before. The results showed that although the extracellular levels of the ePGEs (Figure 2g,i) and levels of SA-βGal rose as D17 approached senescence in the absence of p16^INK4A^ they failed to do so in the cultures treated with ROCKi.

#### 3.2.3. Ectopic *TERT* Expression in the Absence of p16^INK4A^ Bypasses Replicative Senescence and Suppresses the Production of ePGEs

The cell line D17 has been extensively characterized. It lacks p16^INK4A^ expression but possesses functional wild type p53 and normal telomerase regulation [15] plus *NOTCH1* mutation [11]. The ectopic expression of *TERT* restores telomerase activity and bypasses senescence [15]. We therefore compared the levels of ePGEs in senescent D17 with those of D17 TERT cells and showed that ectopic *TERT* expression ablated ePGE1 (Figure 2k) expression and drastically reduced that of ePGE2 (Figure 2l). *TERT* expression also ablated SA-βGal levels although they were not high in the controls (Figure 2m). These data suggest that telomerase can suppress ePGE expression in the absence of p16^INK4A^.

#### 3.2.4. Ectopic *TERT* Expression in the Absence of p16^INK4A^ Is Insufficient to Completely Suppress the Production of ePGEs

Finally, to address the role of senescence effectors in the regulation of the PGEs we compared two cell lines that both originated from the floor of the mouth [14] and had been immortalized by ectopic expression of *TERT* and *CDKN2A*/p16^INK4A^ dysfunction either with (OKF4), or without (OKF6), the inactivation of p53 [14,18]. Figure 1b shows that whilst both ePGE1 and ePGE2 are elevated in OKF6 relative to normal keratinocytes, OKF4 shows normal expression (Figure 1b). The conclusion from these three sets of data is that ePGEs are regulated by telomerase and the senescence program but may also be independently regulated by *TP53* and *NOTCH1*.

### 3.3. MPPOLs Do Not Display Increased Levels of All Senescence Markers but Do Show Reduced Levels of the Licensing Factor MCM2/7 and Increased Levels of Some but Not All SASP Factors

To test whether the MPPOLs were more senescent than the NOKs at the time of ePGE analysis, we performed Western blotting for early and late senescence markers [38]. The results confirmed previously published work on the status of these markers in these cells [7,8,14,18,39]. However, although the MPPOL lines generally showed a reduction in the proliferation markers they did not show any evidence of elevated p16^INK4A^ or reduced levels of sirtuin1, which are the markers of late senescence in fibroblasts [11]. MPPOLs D6, D30 and E4 and OKF6 all displayed several SASP markers such as the increased extracellular IL-6 and IL-8 protein (Appendix A). Levels of IL-8/*CXCL8*, *TGFB2*, *CXCL5*, *CXCL6*, *CCL20* and *MMP2* transcript were also higher in MPPOL cultures, especially in D30 (Appendix A). However, notably, a few others were not higher, such as *IL1A* and *IL1B* transcript or increased extracellular IL-1α and IL-1β protein and transcript (Appendix A). In conclusion, MPPOL keratinocytes were not fully classically senescent at the time of analysis but did display several markers of the SASP, such as increased extracellular IL-6 and IL-8 protein, together with increased transcripts of other cytokines and MMPs, suggesting that MPPOL keratinocytes were more senescent than their normal counterparts. However, MPPOL can still proliferate and do not have elevated levels of p16^INK4A^ [11].

### 3.4. The Upregulation of the ePGEs Is Dependent on Cyclo-Oxygenase 2 (COX2) but Not Cyclo-Oxygenase 1 (COX-1)

Western blotting showed that COX1 was, if anything, reduced in all MPPOL and IPPOL keratinocytes relative to the normal controls (Figure 3a). However, the levels of COX2 were consistently elevated in the MPPOL and the non-neoplastic immortal line OKF6 keratinocytes compared to normal in parallel with the levels of ePGEs (Figure 1 and Figure 2b). In contrast, COX2 was often normal or down-regulated in the immortal IPPOL lines and the non-neoplastic immortal oral keratinocytes line OKF4 in parallel with low ePGE levels (Figure 2b). Both the COX1 and COX2 antibodies were validated by siRNA knockdown in OKF6 cells (Figure 3c). To investigate further the differential regulation of ePGEs and ILs in MPPOL and IPPOL keratinocytes by COX-2 and other pathways, we used OKF6 cells, as they expressed significant levels of both PGEs and the ILs 1, 6 and 8, despite being immortal. MPPOLs grew slowly and were hard to generate in sufficient numbers. We treated OKF6 keratinocytes with salicylic acid (Figure 3d), NS-398 (Figure 3d) and the COX-2 selective inhibitor celecoxib (Figure 3e) and showed that all treatments reduced both PGE1 and PGE2 to baseline levels and these results were highly significant. We also knocked down (KD) COX-1 and COX-2 with siRNA by 44% and 53% relative to the mock transfected controls, respectively (Figure 3c). Notably, only the COX-2 KD gave a consistent 25–35% reduction in ePGE1 and ePGE2 levels in the conditioned media of the OKF6 cells relative to both the mock-transfected and scrambled siRNA controls (Figure 3f). These data are consistent with the celecoxib data, but did not reach statistical significance for ePGE1. In conclusion, these results suggest that COX-2 is the major mediator of ePGE upregulation in OKF6 cells and is responsible for the upregulation of both ePGEs in the MPPOL keratinocytes.

### 3.5. The Upregulation of the ePGEs Is Dependent on p38 Mitogen-Activated Kinase (p38MAPK) a Known Regulator of COX2

It has been reported that p38MAPK can upregulate COX-2 [40,41] and induce senescence [42] and SASP cytokines [43] independently of telomere function. Therefore, we treated OKF6 cells with the p38MAPK inhibitor SB203580 and measured ePGE levels to test the hypothesis that this pathway was responsible for the higher levels of these metabolites in some immortal lines and also to test the requirement for p53. The results showed that SB203580-treated cells expressed much lower levels of ePGEs than the controls (Figure 4a,b) and that COX-2 levels were reduced (Figure 4c). The reduced level of heat shock protein 27 (HSP-27) phosphorylation in the SB203580-treated keratinocytes confirmed that the drug had worked (Figure 4c). These results indicate that the effect of p38MAPK on ePGEs and COX-2 was independent of p16^INK4A^, which is absent from OKF6 and was not completely suppressed by telomerase in this line. ePGEs were largely undetectable in D17 TERT cells, as was COX-2, but senescent D17 did express detectable ePGEs and SB203580 reduced the expression of both ePGE1 (Figure 4d) and ePGE2 (Figure 4e). Interestingly, SB203580 had no effect on D17 TERT cells (Figure 4d,e), indicating that the p38MAPK pathway was compromised in D17 and was over-ridden by ectopic *TERT* expression. 

### 3.6. The DNA Damage Response (DDR) Is Not Specifically Elevated in MPPOL Relative to IPPOL Keratinocytes

As cellular senescence and the SASP have been linked to irreparable DNA double strand breaks [43] located at the telomeres [44,45], we tested the role of the DNA damage response (DDR) proteins in PPOL ePGE regulation. p53 phosphorylation relative to total p53 was consistently elevated (two- to three-fold) in the MPPOL keratinocytes compared to normal (Figure 5a) and was highest in line D30 with the highest levels of ePGEs. However, relative p53 phosphorylation was below normal levels in all the IPPOLs except D4, D17 and the immortal non-neoplastic lines, OKF4 and OKF6, all of which express wild type p53.

The upstream DDR checkpoint kinases ATM and CHK2 have been reported to be increased in certain human pre-malignancies in vivo [46]. ATM showed a consistent upregulation in total protein (Figure 5b) in all PPOLs and OSCCs and has been reported following DNA damage before [47]. However, only one MPPOL line, D25, showed an increase in phosphorylation at serine 1981 (ATM-P) relative to NOKs. Therefore, although the increased p53 phosphorylation in the MPPOL keratinocytes was not mediated through ATM-P, the consistent increase in total ATM in MPPOL keratinocytes could be indicative of increased DNA double strand breaks. Furthermore, ATM-P remained very high in D17 and all IPPOL and immortal OSCC (IOSCC), suggesting that ATM-P was not enough to induce ePGEs. ATM-P has recently been reported to be essential for the addition of telomere repeats in telomerase expressing cells [48] and as all IPPOL [7] and OSCC [20] lines have deregulated telomerase, ATM may not just serve a DDR function in these cells. In support of this hypothesis, the ATM inhibitor KU55933 had no detectable inhibitory effect, even in OKF6 (Figure 5c) or senescent D17 cells (Figure 5d), and if anything showed a slight trend towards the stimulation of ePGEs in senescent D17 cells (Figure 5d).

We also examined the status of CHK2, but there was no striking variation of either total or phosphorylated CHK2 in the cell line panel (Appendix A), suggesting that this kinase was not responsible for ePGE upregulation in MPPOL keratinocytes. In conclusion, whilst there was evidence for an increase in p53 phosphorylation in MPPOL relative to NOKs this is not explained by phosphorylation of the DDR checkpoint kinases. 

Finally, we tested whether the induction of senescence by the introduction of irreparable DNA damage using ionizing radiation in human epidermal keratinocytes was enough to induce the secretion of ePGEs, but this was found not to be the case either (Appendix A). These results suggest that activation of the DDR and senescence in the MPPOL keratinocytes is insufficient to cause the secretion of ePGEs and that the MPPOL are not merely senescent normal keratinocytes. These data and that presented elsewhere suggest that other signalling pathways, such as p38MAPK, are required in concert.

### 3.7. Regulation of the PGEs by p53

Figure 6 shows that whilst both ePGEs (Figure 6a) are elevated in OKF6 relative to normal keratinocytes, OKF4 shows normal expression, indicating that the PGEs might be regulated by p53. OKF6 also showed abnormally high levels of IL-1α and IL-1β, IL-6 and IL-8 that were absent from OKF4 (Figure 6b). The unusual IPPOL line D4 also had high levels of both the ePGEs and IL-1s, as well as one allele of wild type p53 [7], so a role for p53, independent of telomerase and p16^INK4A^, was suggested by these data.

#### p53 Is Necessary for the Upregulation of ePGE1 but Not ePGE2 in OKF6 Cells

To test the role of p53 in PGE regulation, we knocked down p53 in OKF6 cells using siRNA and confirmed the level of knockdown as between 60 and 80% by Western blotting (Figure 6c). Conditioned medium was collected from the p53 knock-down (p53KD) cells and the levels of ePGE1, ePGE2, IL-6 and IL-8 (Figure 6d) were measured. The results showed that ePGE1 but not ePGE2 required p53 for its upregulation in OKF6 cells and IL-6 and IL-8 were both increased slightly by p53 knock-down, consistent with previous reports that p53 restrains IL-6 and IL-8 expression in IrrDSB-induced senescence in fibroblasts [43].

### 3.8. PPOL Keratinocyte Senescence in the Absence of p16^INK4A^ Induces ePGE and a Distinct Set of SASP Factor Transcripts That Are Reversed by Telomerase

To further investigate the role of senescence in the control of the SASP cytokines we mined transcript data [10] from the pseudodiploid *CDKN2A*/p16^INK4A^-deficient D17 cells [7] that had been transduced with the catalytic component of telomerase *TERT* or the empty vector, and were allowed to reach or bypass the senescence checkpoint [15].

Figure 7 shows that transcripts of several of the above cytokines (*CXCL1*, *CCL20* and *IL8*/*CXCL8*) together with *IL1A* and *IL1B* and transforming growth factor α (*TGF**A*) were upregulated in senescent D17 and were reversed upon senescence bypass by the ectopic expression of telomerase, thus linking the expression of these cytokines with telomere dysfunction in the absence of *CDKN2A*/p16^INK4A^. In the same cells, transcripts encoding matrix metalloproteinase (MMPs) and their TIMPs were also similarly regulated with *MMP1*, *9* and *10* being upregulated in senescence and *TIMP1* downregulated (Figure 7). However, *TGFB2*, *CXCL5*, and *CXCL6* levels did not become elevated following D17 senescence and IL-8/*CXCL8* protein levels were not reversed by telomerase in OKF6 cells following extensive passaging, suggesting that extra-telomere signals such as those mediated by p38MAPK/*MAPK14* also regulate SASP cytokines in PPOL keratinocytes, similar to the regulation of ePGEs.

### 3.9. ePGEs Are Not Inversely Related to the Levels of PGE Receptors

One possible explanation for the increased levels of ePGEs in the MPPOL keratinocyte media might be a reduced level of PGE receptors on the keratinocyte surface, thus resulting in less surface binding and increased ePGE in the media. We therefore examined the expression of all four PGE receptors (Eph1-4) in our cell line panel by Western blotting and there was no relationship between PGE receptor level and ePGE expression (Figure 8). D4 and D9 had slightly higher levels of Eph2-4 than D6 but D19 and DOK did not. Linear regression analysis of ePGE levels and EP receptor levels across the cell line panel in the same experiments as the depicted Western blot in Figure 8 showed no correlation between ePGE1 and EP2 (*p* = 0.29; R^2^ = 0.15), EP3 (*p* = 0.32; R^2^ = 0.14) or EP4 (*p* = 0.78; R^2^ = 0.01) or ePGE2 and EP2 (*p* = 0.31; R^2^ = 0.15), EP3 (*p* = 0.27; R^2^ = 0.17) or EP4 (*p* = 0.88; R^2^ = 0.003). Therefore, we consider it unlikely that the high levels of ePGEs in the MPPOL media was due to a reduction of PGE receptors.

### 3.10. The Effect of Hydrocortisone (HC) on the Regulation of Extracellular PGEs

As it has been reported that the doses of steroids routinely used in our keratinocyte cultures suppress the interleukins of the SASP [49] and ePGE2 [50], we tested whether the difference in ePGE levels between MPPOL and IPPOL keratinocytes might be due to differences in their sensitivity to hydrocortisone (HC). We cultured two of the immortal lines, DOK and D19, for 3 weeks in the absence of hydrocortisone. The results (Figure 9) showed that whilst the growth rate slowed and the saturation density decreased in the steroid-deprived keratinocytes, there was only a very modest (two fold) and non-significant increase in ePGE1 (Figure 9a), but there was a 10-fold increase in ePGE2 in D19, which was also non-significant in both D19 and DOK and was not sufficient to stimulate D19 or DOK proliferation (see below). Therefore, although increased sensitivity to HC did not completely account for the large difference in ePGE levels or cytokines between MPPOL and IPPOL keratinocytes [10,11] (see below), it does suggest that MPPOL keratinocytes may regulate ePGEs in a similar manner to the SASP. The results also suggest that MPPOLs have a difference in steroid metabolism from NOKs that may account for some of the results, as even in the presence of HC, MPPOLs express higher levels of ePGEs than IPPOL or normal keratinocytes. However, the effect of HC needs examining in more IPPOL lines.

### 3.11. PGE2 but Not PGE1 Stimulates Proliferation in Some IPPOL Lines Lacking Autocrine Secretion of the PGEs

It has been reported that PGE2 and Eph2 and Eph3 receptor agonists can stimulate the proliferation of some OSCC cell lines [50] and we have extended these experiments to the IPPOL line D19 but not DOK (Figure 9a), suggesting that PGE2 derived from the pre-cancerous environment might stimulate IPPOL cell multiplication as well as OSCC in some cases. The doses used (3–10 ng/mL or 8.5–30.7 nM) are physiological, as D6, D25 and D30 and late passage D17 produce between 5 and 60 ng/mL ePGE2/10^5^ cells (Figure 1 and Figure 2). We found no effect of PGE1 on either line. To test whether there was an autocrine effect of the PGEs on IPPOL cells we treated both D19 and DOK cells with celecoxib. Celecoxib completely eliminated the production of both PGEs in OKF6 cells at a dose of 3 μM and there was no effect on D19 but it actually stimulated DOK slightly at 10 µM (Figure 10b), arguing against an autocrine effect of ePGE2 in IPPOL cells in agreement with the low levels of ePGE in D19 and DOK (Figure 1).

### 3.12. Exogenous PGE2 Does Not Induce Oral Fibroblast Senescence

Senescent oral fibroblasts do not secrete ePGEs [36] but PGE2 has been reported to induce lung fibroblast senescence [51] and therefore keratinocyte-derived ePGE2 could induce the fibroblast SASP. However, doses of 0.85 to 85 nM of both PGE1 and PGE2 had no effect on the frequency of senescent oral fibroblasts as assessed by SA-βGal (Appendix A). These data did not support the hypothesis that the MPPOL keratinocytes could indirectly promote IPPOL progression by inducing fibroblast senescence in the cancer-associated mesenchyme.

## 4. Discussion

Recent and historical data have shown that PPOL and OSCCs are composed of mixtures of mortal and immortal cells [7,8,9] that have strikingly distinct properties and appear to diverge further throughout tumor progression [10]. Paradoxically, the mortal diploid cells express more classical OSCC markers than the immortal ones [10], the latter of which form experimental tumors and contain established cancer driver mutations [12,13]. The nature of the mortal cells in these tumors is not clear; whilst they could be a distinct form of cancer underpinned by microRNA or non-coding RNA changes, it is also possible that they are normal keratinocytes recruited into the tumor to provide a specific role, i.e., ‘cancer-associated keratinocytes’. Some MPPOL lines have also been shown to be resistant to suspension-induced terminal differentiation [8] and to over-express keratin 7 [10], which is associated with glandular epithelium, and so some MPPOL may have stem cell properties.

We show here that ePGE1 and ePGE2 and several cytokines are strikingly upregulated in MPPOL relative to NOKs and this upregulation was not explained by differential regulation of the PGE receptors. ePGE upregulation was associated with COX-2 expression and a subset of SASP factors, but not with the DDR pathways. Upregulation of the ePGEs was not completely reversed by telomerase or inhibitors of the DDR, even in the absence of p16^INK4A^ and inhibition of COX-2 and p38MAPK was also required. In addition, inducing the DDR and senescence in normal keratinocytes was not enough to upregulate ePGEs.

All of these pathways are known to regulate the cytokines, which are part of the fibroblast SASP [38,43,52] and mining data from our previous microarray study [10] showed that some SASP-like proteins, but certainly not all, are similarly upregulated in MPPOL keratinocytes. Furthermore, SASP factor transcripts elevated in MPPOL keratinocytes approaching senescence were different from those following the replicative senescence of a p16^INK4A^-defcient MPPOL line D17, with the exception of IL-8. This suggests that the nature and role that the SASP plays in the development and progression of OSCC is context- and mechanism-dependent.

In addition, ePGE1, but not ePGE2, did show dependence on p53, even in the telomerase-positive OKF6 line, suggesting alternative pathways of ePGE1 regulation but the mechanism underpinning the differential regulation of the ePGEs by p53 is not presently clear.

With few exceptions, IPPOL do not upregulate ePGEs and SASP-related cytokine transcripts and often downregulate them, which may enable them to evade clearance by the innate and adaptive immune systems as published previously in mouse cancer models [26,53]. Some of this resistance may be related to their increased sensitivity to hydrocortisone, as withdrawal of the latter partially restored ePGE production, but this restoration was only partial and not significant.

IPPOL generally show less differentiation in conventional 2D cultures than MPPOL and do not display many important markers of OSCC such as α_v_β_6_ integrin and this protein is absent from undifferentiated OSCC in vivo [54]. These data give rise to the hypothesis that in complex pre-cancerous fields of mortal and immortal keratinocytes the MPPOL keratinocytes produce numerous molecules that have the potential to induce tumor progression of the more advanced IPPOL and OSCC keratinocytes. It is well documented that PGE2 can stimulate the proliferation of a variety of neoplastic cell lines, including head and neck SCCs [50] and we have extended this observation to some IPPOL lines with minimal autocrine ePGE2 secretion. However, there was no evidence that ePGE2 could induce senescence in oral fibroblasts. These results support a paracrine effect of MPPOL keratinocytes on IPPOL multiplication via PGE2.

PGE1 is known to stimulate angiogenesis, another essential process for continued tumor growth in vivo [55,56], and cytokines such as IL-6 and IL-8 have been shown to promote cancer cell invasion [57]. This hypothesis is further supported by recent data showing that in the mouse two-stage carcinogenesis model, the source of PGE2 is dependent on epidermal COX2 [58] and progression to SCC is dependent on both epidermal COX2 [58] and senescent keratinocytes [59].

Although it is accepted that the phenotypes displayed by MPPOL keratinocytes may have occurred or have been enhanced in vitro, there is support for our hypothesis from histological studies in vivo. Notably, although COX-2 expression is upregulated in PPOL and OSCC in vivo, it is mainly expressed in the upper layers of the mucosa [60] and is highly heterogeneous [60], which is consistent with our results. Although there is no reported evidence for senescence in human PPOL lesions in vivo, there is in other human pre-malignancies [46] and p16^INK4A^ is upregulated in high-grade PPOL lesions [61]. Interestingly, mice ectopically expressing p16^INK4A^ in epidermal keratinocytes have recently been reported to express features of senescence and exert paracrine tumor-promoting effects on neighbouring keratinocytes [62]. Therefore, there is evidence that senescence and ePGEs can promote the progression of pre-malignant keratinocytes to OSCC in vivo. However, to test our hypothesis definitively it will be necessary to explore the expression patterns of individual PPOL keratinocytes in vivo in different states of differentiation to test these hypotheses further.

Significantly, regulators of MPPOL ePGEs, cytokines and MMPs such as p38MAPK and COX-2 are established anti-cancer targets and the elimination of senescent cells by senolytic drugs has been shown to inhibit epidermal SCC progression [59], suggesting that the targeting of MPPOL keratinocytes in a pre-cancerous field could be an effective therapeutic and/or chemotherapeutic strategy. Senolytics targeting MPPOL may be required, as cell type specificity of this class of drug has already been observed [63]. Of additional significance is the observation that IPPOL and IOSCC keratinocytes do not generally express significant levels of ePGEs unless hydrocortisone is withdrawn from the culture medium, suggesting that therapies based on such targets could be complemented by strategies that inhibit the steroid receptor. In addition, the induction of IPPOL keratinocyte senescence or differentiation, as previously suggested by others based on a p53-deficient mouse liver cancer model [53], may aid therapeutic effectiveness of the targets mentioned above.

The limitations of our study as it stands is that the differences between NOK, MPPOL and IPPOL keratinocytes are observed in a partially differentiated keratinocyte system, albeit with physiological levels of plasma hydrocortisone. COX-2 [60] and many markers of PPOL and OSCC in vivo [10] are expressed by the MPPOL and not the IPPOL keratinocytes. This is curious given that the latter are generally held to be the precursors of tumorigenic OSCC. Future work should include testing the hypotheses that an IPPOL gene signature exists in PPOL and OSCC tissue samples by single cell RNA sequencing [64] and, in addition, that IPPOL keratinocytes can re-express COX-2, the ePGEs and SASP proteins when placed in 3-dimensional organotypic cultures.

## 5. Conclusions

Our results (summarized in Figure 11), together with recent data from mouse models of epidermal carcinogenesis, support the hypothesis that ePGEs and cytokines can originate from oral keratinocytes approaching senescence in the pre-cancerous field but that the breakdown of senescence ablates their expression. As these molecules are known to promote tumor proliferation, invasion and angiogenesis, our results support the hypothesis that the paracrine delivery of these molecules may be necessary to drive PPOL progression. We also identify p38MAPK, COX2 and steroids as potential anti-cancer target pathways.

## Figures and Tables

**Figure 1 cancers-14-02636-f001:**
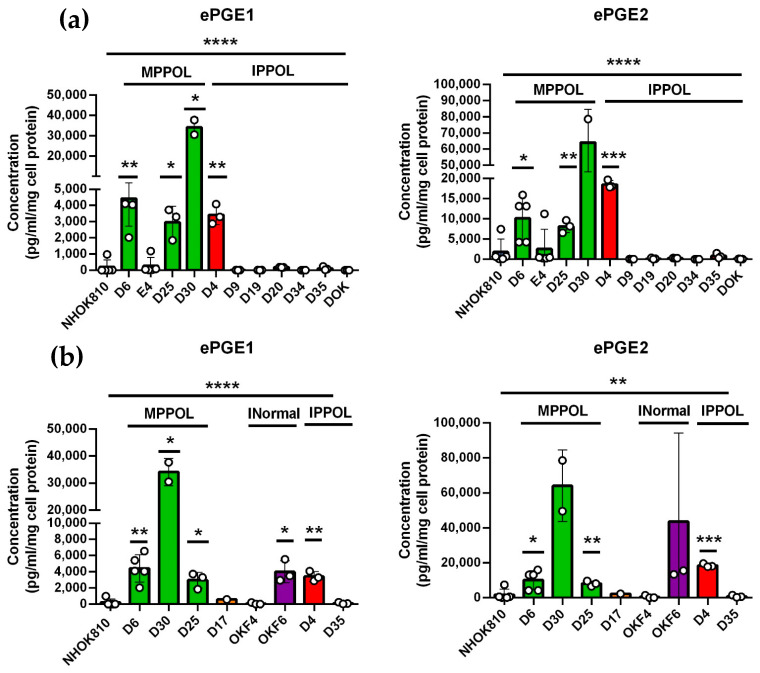
(**a**) The levels of ePGE1 (left panel) and ePGE2 (right panel) in normal (NHOK810 pale blue bars), a subset of the MPPOL (green bars) and IPPOL (red bars) keratinocytes. (**b**) NHOK810 (pale blue bars) compared with the p16^INK4A^−/− MPPOL line D17 (orange bar) at early passage and the normal keratinocyte lines were immortalized with defined genetic elements OKF4 and OKF6 (purple bars) assayed with NHOK810 separately. The data are expressed as the level of ePGE in ng/mL/mg cell protein and was derived from 3 independent experiments (*n* = 3) ± standard deviation assayed in duplicate except for NHOK810 (*n* = 5); D6 (*n* = 5); D30 (*n* = 2); D17 (*n* = 1). * *p* < 0.05; ** *p* < 0.01; *** *p* < 0.001; **** *p* < 0.0001 when compared to the normal keratinocyte line NHOK810. Line over the graphs is an analysis of the data sets by one-way ANOVA. MPPOL = mortal PPOL lines; IPPOL = immortal PPOL lines; INormal normal oral keratinocytes immortalized by defined genetic elements and/or *TERT*.

**Figure 2 cancers-14-02636-f002:**
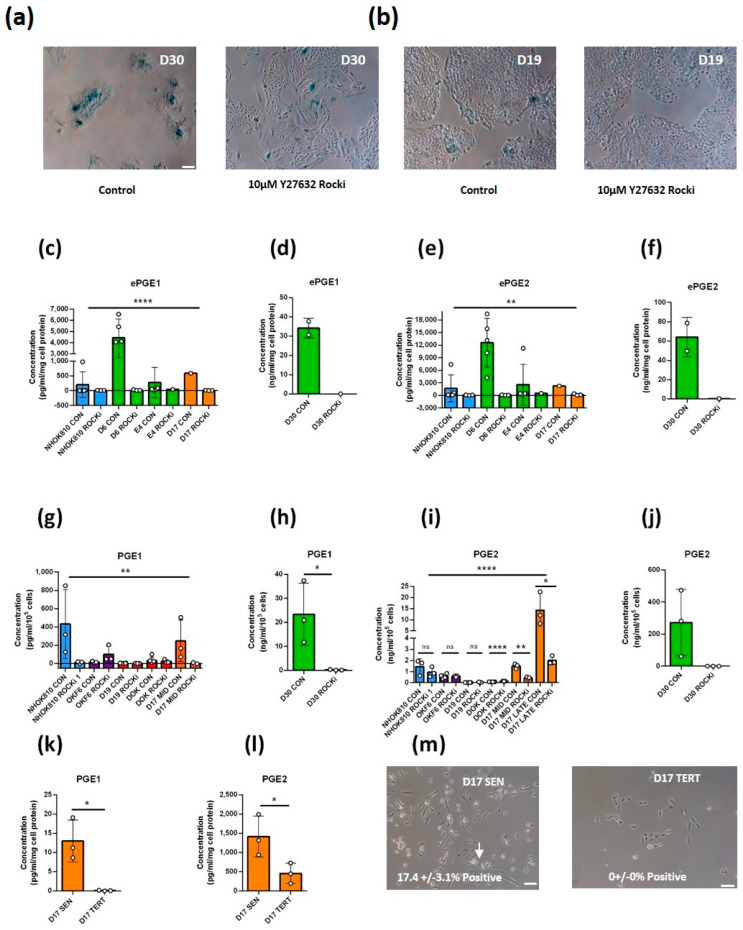
Ablation of the senescence program by ROCKi or ectopic *TERT* expression inhibits the expression of ePGEs. (**a**) SA-β staining of MPPOL line D30 +/− ROCKi that showed reduced staining by ROCKi. Scale bar = 100 µm. (**b**) SA-β staining of IPPOL line D19 as in (**a**) at the same magnification showing essentially no staining of the cells and unaffected by ROCKi. (**c**–**j**) The effect of ROCKi on normal (NHOK810), normal immortal OKF6, MPPOL (D6, D30 and E4), D17 MPPOL p16^INK4A^−/− (at 28 (mid) and 36 (late) mean population doublings). IPPOL lines showing a striking reduction in ePGE1 (**c**,**d**,**g**,**h**) and PGE2 (**e**,**f**,**i**,**j**) in mortal, but not in immortal, keratinocyte lines. ePGE1 (**k**) and ePGE2 (**l**) was strongly suppressed by the ectopic expression of *TERT* in D17 along with SA-β staining scale bar = 100 µm (**m**). The data are means +/− standard deviation; *n* = 3 except for D30 where *n* = 2. The color codes are the same as in Figure 1. The unpaired Student’s *t* test was used to compare two groups (ROCKi versus control) and one-way ANOVA (upper bar in (**c**,**e**,**g**,**i**)) to compare multiple groups * = *p* < 0.05; ** = *p* < 0.01; **** = *p* < 0.0001; ns = not significant.

**Figure 3 cancers-14-02636-f003:**
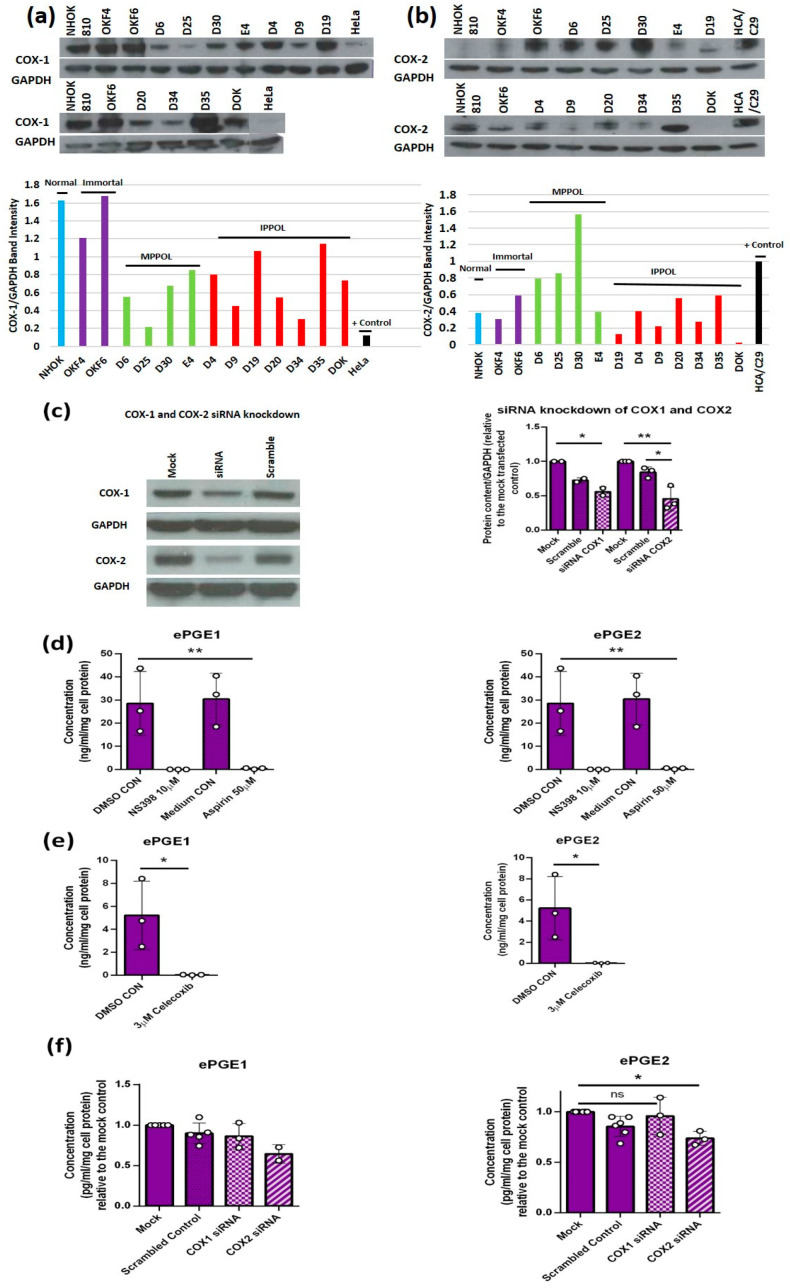
ePGE expression in MPPOL keratinocytes is mediated by COX-2. (**a**) Shows a Western blot (upper panel) and quantitation of the band intensity (lower panel) of COX-1 expression in the cell line panel analyzed in Figure 1 for ePGEs and shows no over-expression of COX-1 in the MPPOL lines. HeLa extract served as a positive control. (**b**) Shows a Western blot and quantitation of COX-2 in the same extracts as in (**a**), showing clear over-expression of COX-2 in MPPOL lines D6, D25 and D30 and OKF6 but not in the IPPOL lines. HCA/C29 extract served as a positive control. (**c**) Shows a representative Western blot of 3 independent siRNA knockdown experiments in OKF6 (left panel) and quantitation of the band intensity normalized to the mock-transfected control in all 3 blots +/− standard deviation * *p* < 0.05; ** *p* < 0.01 by Student’s unpaired *t* test. The data show significant knockdown of COX-2 (striped bar) relative to both the mock-transfected and scrambled controls and COX-1 (dotted bar) relative to the mock transfected control, respectively, thus validating the antibodies used in (**a**,**b**). (**d**) ePGE1 (left panel) and ePGE2 (right panel) levels in OKF6 cells treated with the COX non-selective inhibitors NS-398 and aspirin showing ablation of ePGE expression ** *p* < 0.01 by one-way ANOVA. (**e**) ePGE1 (left panel) and ePGE2 (right panel) levels in OKF6 cells treated with the COX-2 selective inhibitor celecoxib showing ablation of ePGE expression * *p* < 0.05 by Student’s unpaired *t* test. (**f**) ePGE1 (left panel) and ePGE2 (right panel) levels in OKF6 cells following siRNA knockdown of the cultures characterized in (**c**) normalized to the mock-transfected controls * *p* < 0.05 by Welch’s *t* test; mock-transfected and scramble controls (*n* = 6), Cox-1 hatched bar; *n* = 3) and COX-2 (striped bar; *n* = 3).

**Figure 4 cancers-14-02636-f004:**
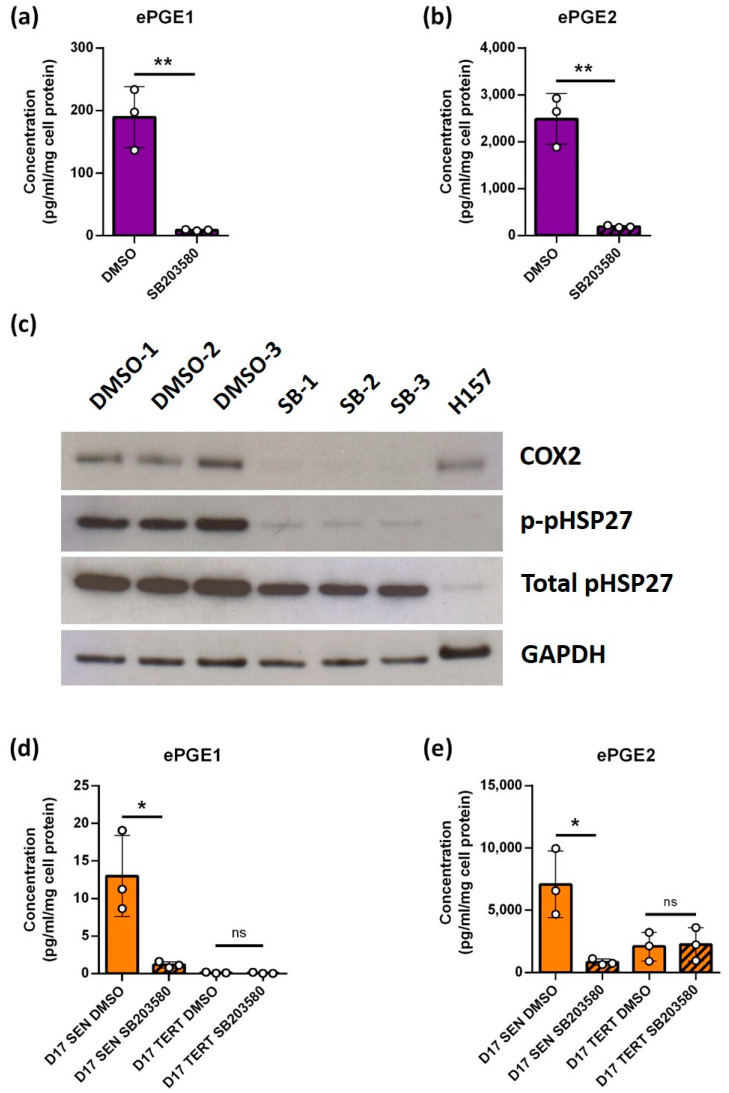
ePGE expression in OKF6 keratinocytes is mediated by p38MAPK. (**a**,**b**) show ePGE1 and ePGE2 levels in OKF6 cells treated with the p38MAPK inhibitor SB203580 (striped bar) along with the DMSO control (plain bar). The figure shows a dramatic reduction in ePGE1 (**a**) and ePGE2 (**b**) following treatment in 3 independent OKF6 cultures (+/− standard deviation, ** *p* < 0.01 by Student’s unpaired *t* test. (**c**) shows representative Western blots of all three independent cultures indicating a dramatic reduction of COX-2 and heat shock protein (HSP-27) phosphorylation (p-pHSP-27) following treatment of OKF6 cells with SB203580 (SB 1-3). GAPDH was used as a loading control and H157 as a positive control for COX-2. (**d**,**e**) Show similar data from senescent D17 cells (D17SEN) and the same cells immortalized by ectopic expression of the catalytic subunit of telomerase *TERT* (D17 TERT) for ePGE1 (**d**) and ePGE2 (**e**). SB203580 (striped bars) inhibited both ePGEs in D17 SEN, * *p* < 0.05 by Student’s unpaired *t* test but had no effect in D17 TERT.

**Figure 5 cancers-14-02636-f005:**
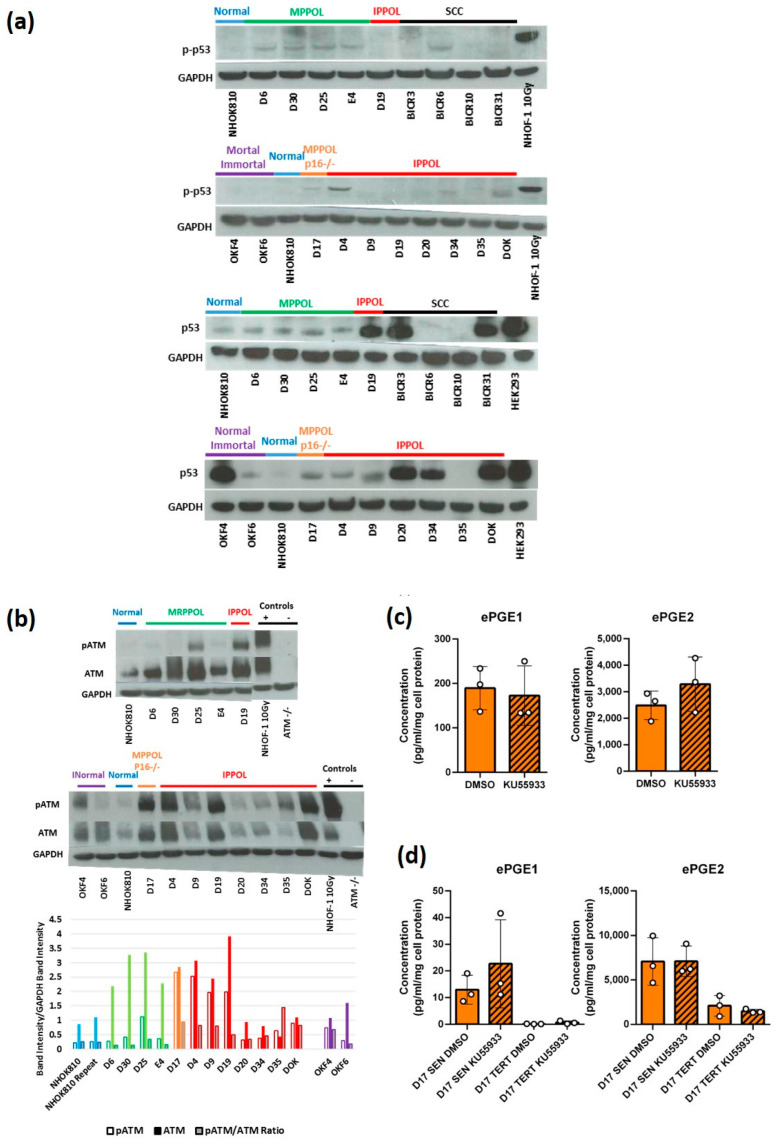
Elevated DDR signalling is not essential for the elevation of ePGEs in MPPOL. (**a**) Shows representative Western blots of serine 15 phosphorylated p53 levels (pp-53) (top two panels). Normal (NHOK810 both panels), MPPOL (D6, D25, D30 and E4 top panel), normal immortalized (OKF4 and OKF6 (second panel), IPPOL lines D19 (both panels) D4, D9, D20, D34, D35 and DOK second panel), MPPOL line D17 lacking p16^INK4A^ (second panel) and p53-characterized OSCC lines BICR3, BICR6, BICR10 and BICR31 (top panel). NHOF-1 oral fibroblasts irradiated with 10 Gy gamma rays served as a positive control. The same extracts run on a separate gel were re-probed for total p53 (panels 3 and 4 respectively) and HEK293 extract served as a positive control. All the p53 blots were re-probed with GAPDH antibody as a loading control. (**b**) Representative Western blots of ATM phosphorylated at serine 1981 (ATM-P) and total ATM (ATM) (NHOK810 both panels), MPPOL (D6, D25, D30 and E4 top panel), normal immortalized (OKF4 and OKF6 second panel), IPPOL lines D19 both panels D4, D9, D20, D34, D35 and DOK (second panel), MPPOL line D17 lacking p16^INK4A^+/− (second panel). NHOF-1 oral fibroblasts irradiated with 10 Gy gamma rays served as a positive control and normal human keratinocytes deficient in ATM served as a negative control. (**c**) ePGE levels in OKF6 cells treated with the ATM kinase inhibitor KU55933 (striped bars) for 72 h +/−standard deviation. (**d**) ePGE levels in senescent (SEN) and *TERT*-immortalized D17 cells treated with the ATM kinase inhibitor KU55933 (striped bars) for 72 h +/− standard deviation. The results (**c**,**d**) were not significant by Student’s unpaired *t* test.

**Figure 6 cancers-14-02636-f006:**
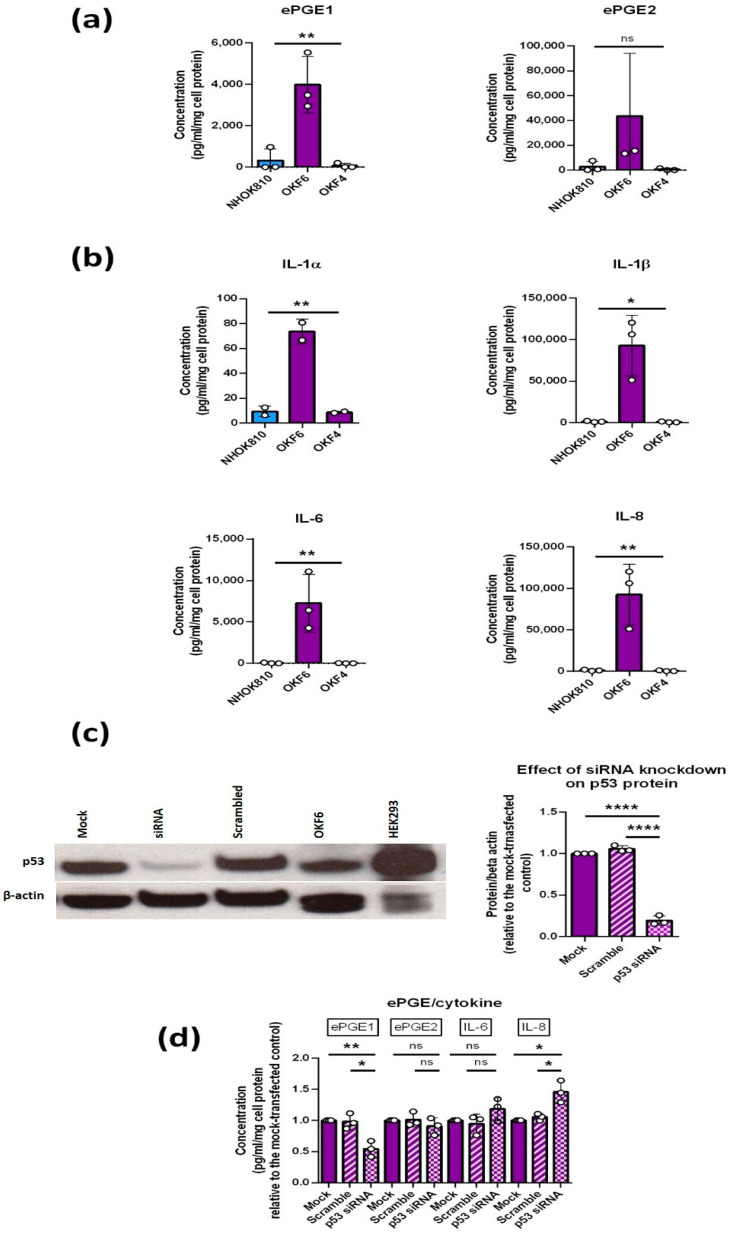
ePGE and cytokine regulation in OKF6 keratinocytes by p53. (**a**) Shows ePGE1 levels and ePGE2 levels in normal (NHOK810), OKF6 (p53 wild type) and OKF4 (p53 dysfunctional) oral keratinocytes and (**b**) shows cytokine IL-1α, IL-1β, IL-6 and IL-8 levels in the same lines as (**a**). (**c**) Shows a representative Western blot of p53 levels in 3 independent siRNA experiments and the quantitation and reduction of band intensity relative to the β actin control +/− standard deviation **** *p* < 0.001 by Student’s *t* test. (**d**) The effect of TP53 knockdown on ePGE1, ePGE2, IL-1α, IL-1β, IL-6 and IL-8 levels in the experiment are validated in (**c**). The data show that only ePGE1 levels were positively regulated by p53 in OKF6, whilst IL-6 and IL-8 levels were restrained by p53 as reported previously in senescent fibroblasts [43]. * *p* < 0.05; ** *p* < 0.01 by Student’s *t* test. Scrambled control RNA (striped bars); p53siRNA (dotted bars).

**Figure 7 cancers-14-02636-f007:**
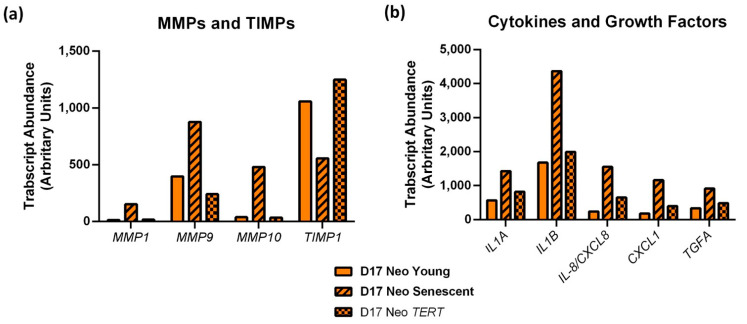
Senescent p16^INK4A−/−^ deficient D17 cells (Striped bars) show elevated SASP factors regulated by telomerase (Dotted bars). (**a**) Shows *MMP* and *TIMP* transcripts regulated by senescence and telomerase in D17 and (**b**) shows the cytokine and growth factor transcripts regulated in the same experiment as (**a**). Data mined from the microarray study in reference [10].

**Figure 8 cancers-14-02636-f008:**
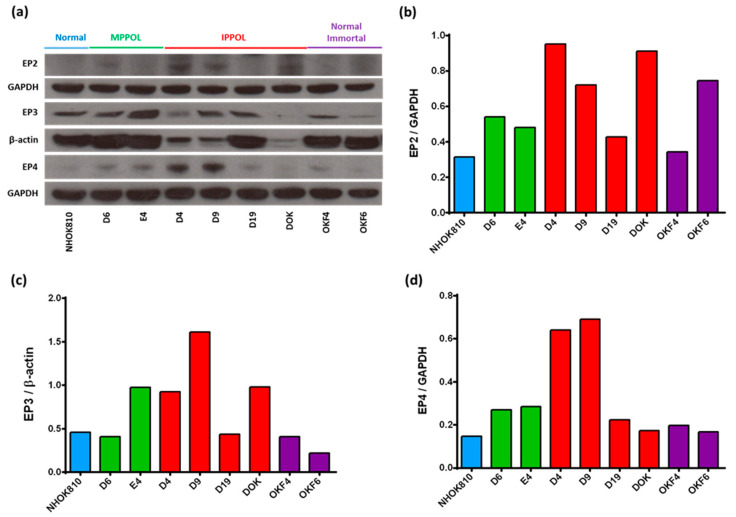
(**a**) Western blot showing the levels prostaglandin receptors (Eph 2-4; EP2, EP3 and EP4 in a selection of MPPOL, IPPOL and NOKs (Eph1 was not detected). GAPDH and beta actin were used as loading controls. (**b**) Quantitation of the EP2 levels in (**a**). (**c**) Quantitation of the EP3 levels in (**a**). (**d**) Quantitation of the EP4 levels in (**a**).

**Figure 9 cancers-14-02636-f009:**
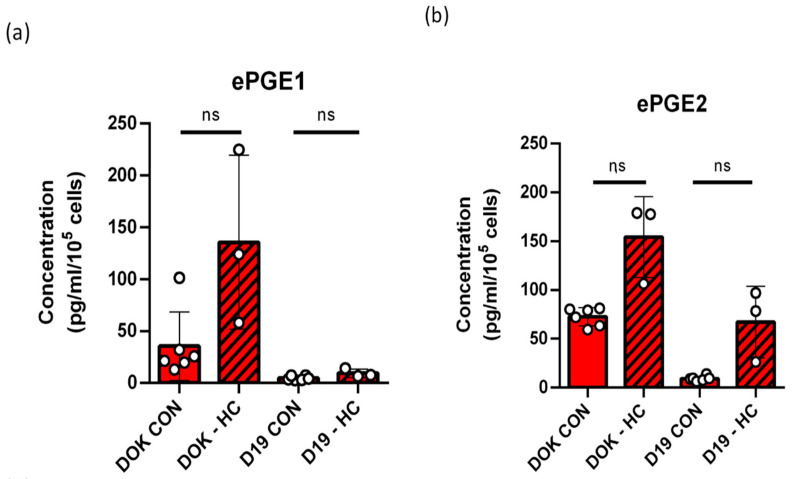
ePGE expression in IPPOL keratinocytes is repressed by hydrocortisone. (**a**) ePGE1 levels and (**b**) ePGE2 levels in DOK and D19 cells where hydrocortisone (HC) was removed from the culture medium for 3 weeks (−HC striped bars). ns = not significant by Welch’s *t* test. CON = control (*n* = 6 independent cultures); −HC = derived of hydrocortisone for 3 weeks (*n* = 3 independent cultures).

**Figure 10 cancers-14-02636-f010:**
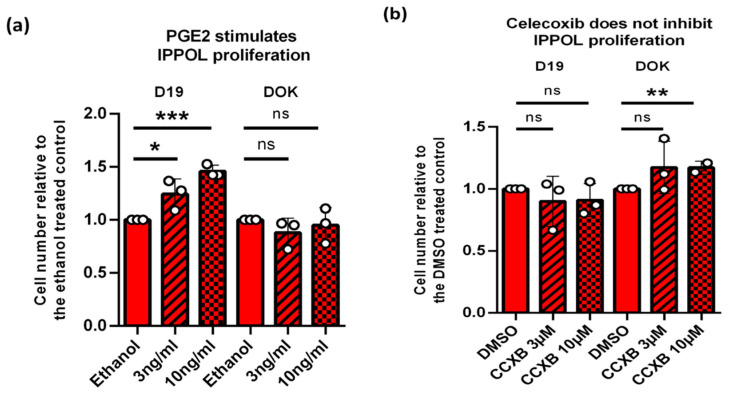
Exogenous PGE2 (3 ng/mL striped bars; 10 ng/mL dotted bars) stimulates the proliferation of some IPPOL lines. (**a**) Shows the effect of a 10-day treatment of IPPOL lines D19 and DOK relative to the ethanol vehicle control +/− standard deviation. (**b**) Shows the effect of the COX-2 selective inhibitor celecoxib (CCXB) on the same lines at 3 µM (striped bars) and 10 µM (dotted bars). * *p* < 0.05; ** *p* < 0.01; *** *p* < 0.001 by Student’s *t* test. The data show that only exogenous PGE2 stimulates D19 proliferation.

**Figure 11 cancers-14-02636-f011:**
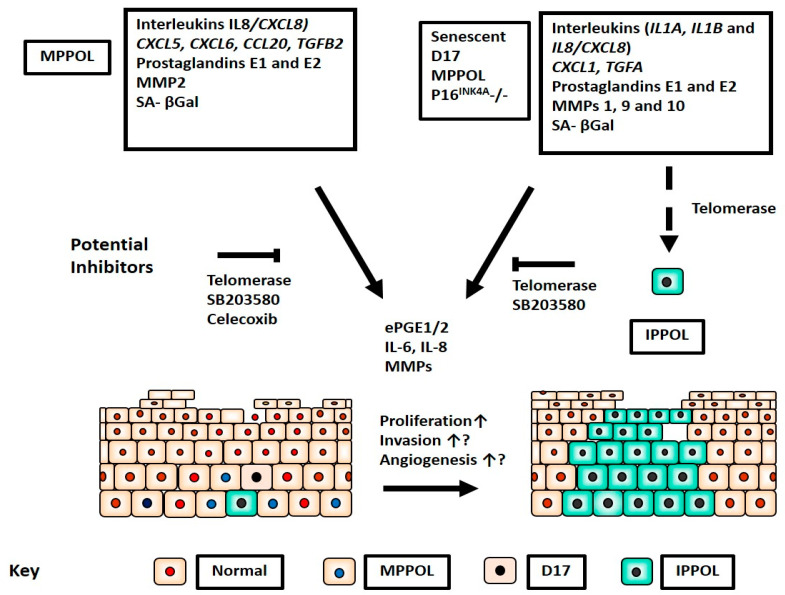
Summary of ePGE and SASP regulation in PPOL keratinocytes, their potential functions in a pre-cancerous oral field and therapeutic targets. The figure summarises the potential impact of the new findings in the context of the different keratinocyte types in a pre-cancerous oral field of largely normal oral keratinocytes (red nuclei; beige cytoplasm). It hypothesises that genetically normal MPPOL keratinocytes (blue nuclei; beige cytoplasm) and senescent keratinocytes en route to immortality lacking p16^INK4A^ such as D17 (black nuclei; beige cytoplasm) collectively create a tumor-promoting environment for IPPOL (black nuclei; green cytoplasm).

## Data Availability

(A) Materials: All cell lines will be made available subject to a reasonable request and the demonstration that the receiving laboratory has the means to maintain the cell lines successfully. Many lines are mortal but can be maintained using a 3T3 feeder layer and the Rho kinase inhibitor Y27632 as described in the methods section. The cell lines will be banked should funds become available. (B) Data sharing: All the raw data files used in the original metabolomic screen will be made available on request. Unfortunately, Metabolon data are not acceptable for deposition in sites such as Metabolytes, as the company will not provide the necessary technical details.

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
