# Peer review of "Extracellular Prostaglandins E1 and E2 and Inflammatory Cytokines Are Regulated by the Senescence Program in Potentially Premalignant Oral Keratinocytes"

_cancers, 2022, doi:10.3390/cancers14112636_

Round 1

Reviewer 1 Report

The manuscript is well written and focuses on an interesting topic.

However, there is minor remarks.

1) In the first sentence of the abstract is “potentially pre-malignant lesions (PPOLs)”. Is the word “oral” missing and should it be “potentially pre-malignant oral lesions?

2) In the abstract the “Material and methods” –section should be described more profoundly including the number of the samples.

3) The first sentence of the Introduction is without a reference.

4) The text should be carefully checked that all abbreviations are named when using first time

5) The Introduction includes a lot of information but the authors may consider to make the aim of the study a little bit clearer in the end of the Introduction. The reason (aim) to make the present study should be clearer.

6) In the Materials and methods the statistical analyses should include description of which p-value was considered as statistically significant, p <0.05?

7) The quality (resolution) of the figures could be a little bit better.

Reviewer 2 Report

The paper by Lee Peng Karen-Ng et al, show that MPPOL, but not IPPOL keratinocytes upregulate the release of extracellular tumor promoting cytokines (e.g. interleukins 6 and 8) and prostaglandins E1 and E2 relative to normal oral keratinocytes. Authors, and others, have previously published the impact of specific metabolites detectable in the saliva of oral cancer patients which could be eventually helpful for detecting oral cancer development earlier; therefore, the findings herein are an incremental extension of these observations. This is an interesting study with translatable applicability for oral cancer. The paper is well written, the text is clear and easy to read and the conclusions consistent with the evidence and arguments presented. That’s why it should be accepted for publication in Cancer upon revisions.

Minor comments:

Fig 1: please remove the label ‘Cell line’ from the graphs,

Fig 2: WB should be showed differently reporting the protein marker. Should be repeated at least twice in order to report a statistical significance of the quantification in plots b) c) and d).

Fig 4: WB a) need to be repeated including the Hela cells on the same blot. Report the protein marker, as mentioned for Fig 2 (and for all the figures showed in the article). The plots COX1 and COX2 are missing od the y axes.

Fig 8: WB plot b) controls should be included on the same blot, please repeat it, and also the pATM and ATM, the GAPDH is not acceptable for publication.

I would personally try to combine or move some figures like Fig 9 and 10 in the supplementary material. Also please increase the quality of the graphs presented (some x and y labels are misplaced). In order to be accepted for publication authors need to provide original WB and row data of the Elisa assays performed.

Reviewer 3 Report

This study has used a series of cell culture experiments to investigate roles of prostaglandins PGE1 and PGE2 in keratinocytes of potentially premalignant lesions (PPOLs). A strength of the experiments is the inclusion of multiple cell lines to represent mortal and immortal cells of PPOLs, however presentation of the data is complicated and labelling of figures is at times confusing. Important experiments are frequently performed using single cell lines rather than multiple cell lines of a particular phenotype and the lack of consistent use of cell lines between experiments appears contrived (were cell lines selected because they more strongly supported or were the only cell lines that supported the hypotheses of the authors?). Presentation of the experiments also does not seem to follow a logical order but appears quite haphazard in places. I feel that the study is worth publishing as it adds a considerable amount of new data to the field. However, I strongly suggest that the manner in which the manuscript is structured is modified in order that the underlying hypotheses of the authors and the findings of the study are more clearly enunciated.

  1. To clarify how the multiple experiments performed in this study are associated with our current knowledge of cell types and cellular processes associated with PPOLs and the progression to malignancy of these lesions, it is suggested that the authors include a cartoon of these cells/processes as a reference. This will allow readers to orient the experiments performed and the results derived using these processes. The authors may also wish to insert a second similar image that includes the new information derived in this study.
  2. The explanation of the cell lines in figures needs to be improved by inserting additional labels, colour coding or spacing between bars. There are multiple cell types, either mortal, immortal, genetically modified, etc, and results are graphed against different cell lines. In its present format, it is too difficult to discern these aspects of the experiments and results in some of the graphs (additional labelling has been included in a proportion of the figures).
  3. Presentation of some of the graphs could be improved as in some graphs, a high value in one of the measurements results in all other bars in the graph being indistinguishable (examples include 3c and 3i, but there are multiple other instances). A suggestion would be to use a broken y-axis.
  4. Please check for formatting in graphs for the final version of the manuscript (e.g. Figure 4b).
  5. The labelling of some of the graphs is unhelpful. For example, the graphs in Figure 4f are labelled “siRNA knockdown ePGE1/ePGE2” and the y-axes are simply labelled as “Concentration…”. This makes it appear that PGE levels were knocked down, however, the “Concentration” is actually referring to ePGE1/2 levels and it is COX1/2 levels that are being knocked down. Please re-examine all graphs to ensure that the y-axes in particular specifically indicate what is being measured, and that the graph titles (if included) are not misleading.
  6. There is a difference between “statistical significance” and “biological significance”. For example, in Figure 6a, the ~3-fold higher ePGE1 levels in DOK cultures and ~10-fold higher PGE2 levels in D19 cultures following removal of hydrocortisone from the culture medium are apparently statistically not significant, potentially in part due to the low number of experimental replicates. But are these types of differences and the actual PGE levels expected to be biologically significant? Also, what happens when hydrocortisone is reintroduced to these cultures?
  7. Addition of a brief summary of the limitations of the present study to the Discussion would clarify the findings of the present study and the next steps that will be required to confirm these results beyond cell culture. They would also better contextualise some of the cell culture phenomena (e.g. effects of hydrocortisone withdrawal from the cell culture medium, inconsistent results between cell lines representing the same phenotype).

Round 2

Reviewer 3 Report

The authors have addressed many of the reviewers’ comments and the amended manuscript is easier to follow. A summary diagram of the pathways involved would certainly make the manuscript more accessible. (The authors comment that they have included such a diagram as a graphical abstract, which cannot be duplicated in the body of the manuscript. However the graphical abstract figure is extremely complicated and is not restricted to the investigations carried out in this study. Most readers would not be accessing the graphical abstract as a tool to understand the manuscript itself, so a simple cartoon that places results of this study in context with each other and with the relevant cell type could be useful. Note that this is not a requirement for publication – it is just a suggestion). I am unable to provide outright support for publication of a manuscript where the reproducibility of data does not conform to scientific standards, however the journal editor may allow its publication as the affected experiments do not form a central finding of the research. Several minor presentation errors are noted.

  1. There may be an error in the presentation of graphs in 2c, e and g (y-axis). For example, the y-axis meets the x-axis at -500 in Figure 2c, and all data are depicted to originate at -500. (Same comment but -3000 in Figure 2e).
  2. Please add the size of the magnification bars in figures 2a, b and m to the figure legend. Please add an explanation for the abbreviation ‘ns’ to the figure legend.
  3. In Figures 2c, 2e and 2g, did the effects of ROCKi treatment on PGE expression reach significance for any of the cell lines (contrast the depiction of results in Figure 2i where significant and non-significant results are depicted)? What comparisons are referred to in relation to the bars depicting statistical differences in figures 2c, e and g (and the equivalent bar in figure 2i)?
  4. Data for “D17 LATE” appear to be missing from Figure 2g.
  5. There is a formatting error in Figure 3a (graph).
  6. In Figure 3c-f, the “striped bar” and “dotted bar” are not visible. Colour has been added to the image, but the colour used is too dark to see detail.
  7. In Figure 5a, phosphorylated p53 levels for D30 do not appear to be exceptionally high, as reported in the text. Do the authors have an alternative image to illustrate this (or else modify the text description in line 436 (“especially high”)).
  8. Suggestion: In relation to Figure 9, an additional phrase in the text contrasting the ~2-fold (non-significant) differences in ePGE expression (following hydrocortisone removal from the culture medium) with ePGE levels that are several orders of magnitude higher in MMPOL cells would assist in interpreting the results. (The lack of a suitable number of replicates is very noticeable in this figure).
  9. There are formatting errors in the supplementary data. Part of Figure S2 is cut off (and there are others). If the supplementary data file was saved as a pdf, this may prevent unexpected formatting errors.

Minor grammatical errors

  1. Line 345: “Elevated” is redundant.
  2. Line 573: “therefore” is redundant.
  3. Please use the Greek symbols for (TGF) alpha and beta.
